# Vitellogenesis in Blue Gourami is Accompanied by Brain Transcriptome Changes

**Gad Degani [1,2], Amir Alon [2,3], Akram Hajouj [2] and Ari Meerson [1,2,*]** 

[1]   MIGAL–Galilee Research Institute, POB 831, Kiryat Shmona 1101602, Israel; gad@migal.org.il
[2]   Faculty of Sciences, Tel-Hai Academic College, Upper Galilee 1220800, Israel; amiralon83@yahoo.com (A.A.);
      ak.ram.1995@hotmail.com (A.H.)
[3]   Ma'abarot fish farm, Kibbutz Ma'abarot 4023000, Israel
*   Correspondence: arim@migal.org.il

**Abstract:** The blue gourami (*Trichogaster trichopterus*) is a model for hormonal control of reproduction in Anabantidae fish, but also relevant to other vertebrates. We analyzed the female blue gourami brain transcriptome in two developmental stages: pre-vitellogenesis (PVTL) before yolk accumulation in the oocytes, and high vitellogenesis (HVTL) at the end of yolk accumulation in the oocytes. RNA sequencing of whole-brain transcriptome identified 34,368 unique transcripts, 23,710 of which could be annotated by homology with other species. We focused on the transcripts showing significant differences between the stages. Seventeen and fourteen annotated genes were found to be upregulated in PVTL and HVTL, respectively. Five nuclear transcripts, three of which contain the homeobox domain (*ARX, DLX5, CERS6*), were upregulated in PVTL. Additionally, several receptors previously known to be involved in reproduction were identified, and three of these, G-protein coupled receptor 54, Membrane progesterone receptor epsilon, and Gonadotropin-releasing hormone II receptor (*GPCR, mPR,* and *GnRHR*) were measured by quantitative RT-PCR in brain, pituitary, and ovary samples from PVTL and HVTL stage females. Of these, *GPCR* was highly expressed in the brain and pituitary as compared to the ovary in both PVTL and HVTL. *GnRHR* was highly expressed in the ovary compared to the brain and pituitary, and its levels in the brain were significantly higher in PVTL than HVTL. Brain *mPR* mRNA levels were likewise higher in PVTL than HVTL. In conclusion, this study details changes in the female blue gourami brain transcriptome through yolk accumulation in the oocytes and identifies key genes that may mediate this process.

**Keywords:** Anabantidae; hormone receptors; Trichogaster; vitellogenesis; yolk accumulation

## 1. Introduction

In fishes as in other vertebrates, the brain is the main organ controlling gonadal development, and specifically oogenesis, by the so-called brain, pituitary and gonad (BPG) axis of hormones [1]. Accumulation of yolk in the oocytes at the vitellogenesis (VTL) stage is a crucial process for synthesizing vitellogenin in the liver [2]. Yolk accumulation is controlled by 17β-estradiol ($E_2$), which is part of the BPG in blue gourami females [1,3–6], as described in other fishes, reviewed by [2]. The brain controls the BPG by releasing the gonadotropin-releasing hormone (GnRH) [7], pituitary adenylate cyclase-activating polypeptide (PACAP), and its related peptide (PRP) [8]. In teleosts as in other vertebrates, Kisspeptins (Kiss) have recently received considerable attention as potential key players in the brain as neuroendocrine controllers independent of reproduction [9]. Kiss1 controls the BPG axis, acts on the caudal hypothalamus, and may affect GnRH signaling [10–12]. We recently reported that the relative levels of Kisspeptin 2 (Kiss2) and Kisspeptin receptor (GPCR or Kiss2r) varied significantly between the brains of PVTL and HVTL blue gourami females [13]. These control the release of pituitary

gonadotropins, the follicle-stimulating hormone (FSH), and the luteinizing hormone (LH), which in turn control gametogenesis [2,14].

Many genes involved in the brain control of oogenesis via the BPG axis are affected by the sexual behavior of the male blue gourami [15]. Ovary development includes different stages, including PVTL and VTL (which includes sub-stages LVTL and HVTL), whereas oocyte maturation, ovulation, and spawning occur only during male courtship. The mature female has an asynchronic ovary, where all stages of oocyte development occur both before and after breeding [16]. The exclusive presence of the chromatin nucleolar stage occurs only in juvenile females (PVTL) [17]. The development of eggs (oogenesis) starting with oogonia (the precursors for oocytes), which are characterized by their small size and the presence of only one nucleus, was described in detail [17]. The BPG hormones involved in controlling oogenesis in blue gourami were also studied extensively, reviewed by [1]. Many factors apparently influence oogenesis, including environmental factors, pheromones (steroid glucoronides) [18] and brain-pituitary hormones. Among the oogenesis-mediating hormones affected by environmental factors are: GnRH1 and BnRH3 [7]; adenylate cyclase-activating polypeptide-related peptide (PACAP and PRP) [8], which act on LH and FSH [3,7,15,19]; growth hormone (GH) [20,21]; and prolactin (PRL) [22]. Plasma $E_2$ and testosterone (T) increase during vitellogenesis (VTL), and $17\alpha$, $20\beta$-dihydroxy-4-pregnen-3-one (17,20P) increase during maturation and ovulation in blue gourami females [1,3–6]. An important transcriptome variation during behavior was found in other fish, e.g., zebrafish (*Danio rerio*) [23]. In zebrafish liver, 1046 transcripts are expressed differentially through vitellogenesis [24]. However, no whole transcriptome study has so far revealed the developmentally regulated genes in the brain through the transition to VTL in labyrinth fish.

The aim of this study is to examine the brain transcriptome in female blue gourami, a model for Anabantidae fish, during VTL. We also examine the mRNA levels in the brain, pituitary, and ovary (BPG axis) of several genes involved in reproduction and growth.

## 2. Results

### 2.1. RNA-Seq

The general scheme of the study is presented in Figure 1A.

We prepared two cDNA libraries from pooled brain RNAs of five juvenile and five mature blue gourami females at two stages of oogenesis (PVTL and HVTL, respectively; Figure 1B,C).

Reads of 62.5M and 54.2M, respectively, were obtained from the two libraries after trimming and QC. The combination of both pools served to assemble the transcriptome, while the stage-specific pools were compared to determine differential expression of specific transcripts. Of the 34,368 unique transcripts identified, 23,710 could be annotated by similarity to genes in other species. The list of species included mammalian (*Homo sapiens, Rattus norvegicus, Sus scrofa, Mus musculus*), amphibian (*Xenopus laevis*), and fish (*Danio rerio, Ovis aries,* and *Cyprinus carpio*). Seventeen and fourteen genes annotated by similarity to other species were marked as being significantly ($P < 0.001$) upregulated in PVTL and HVTL, respectively, by analysis software. (This p-value threshold was chosen arbitrarily, as it is not feasible to perform statistical comparisons between two pooled samples.) These are shown in Figure 2A, and their putative functions according to GenBank are summarized in Supplementary Table S1.

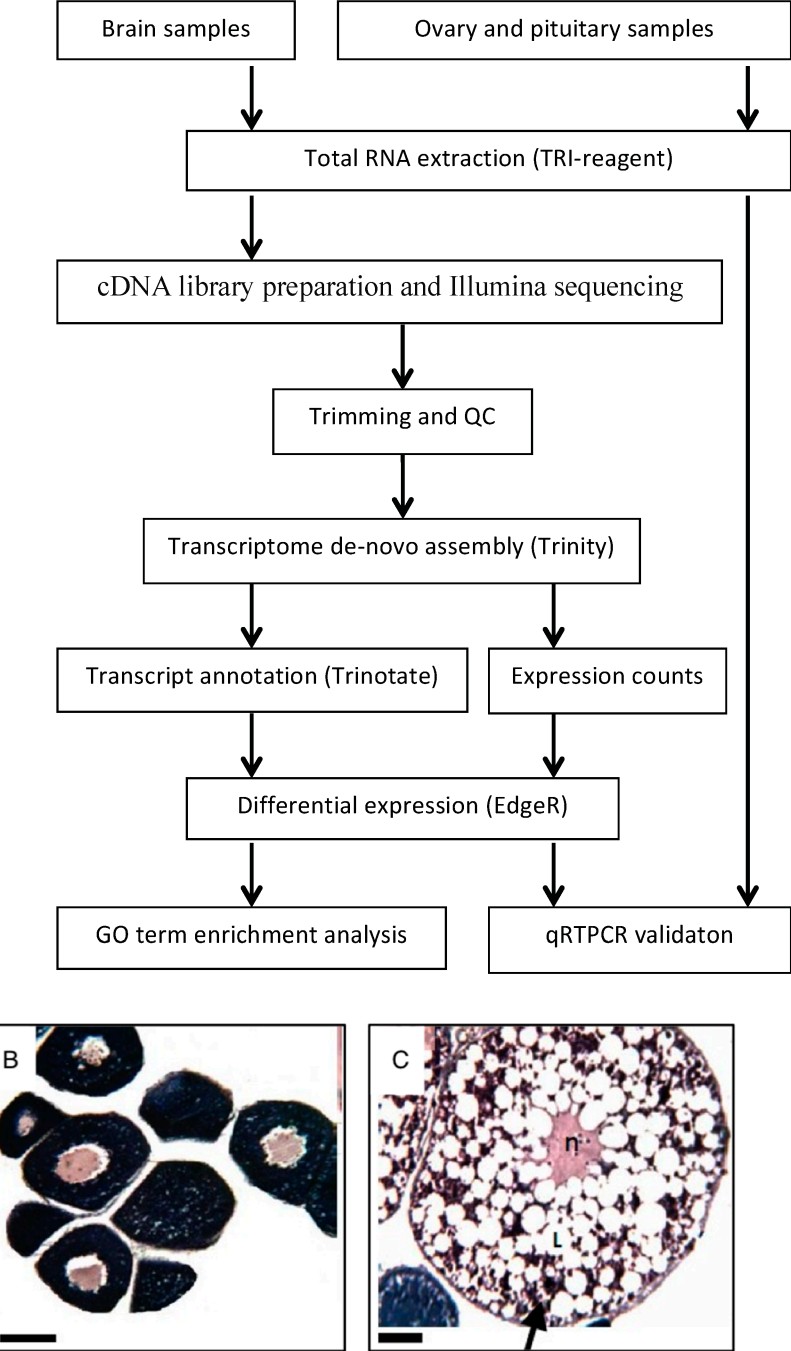

**Figure 1.** (**A**). General scheme of methods used in the present study. (**B**,**C**). Histology of gonadal samples from female blue gourami according to gonadal development stage. (**B**) Young females before vitellogenesis (PVTL). (**C**) Females at advanced stages of vitellogenesis (HVTL). Note the high concentration of yolk vesicles supplanting the periphery of the oocyte (black arrow) and the lipid vesicles (L). The oocyte nucleus is denoted by (n). Sections were stained with hematoxylin and eosin. Bar = 100 μm.

**A**

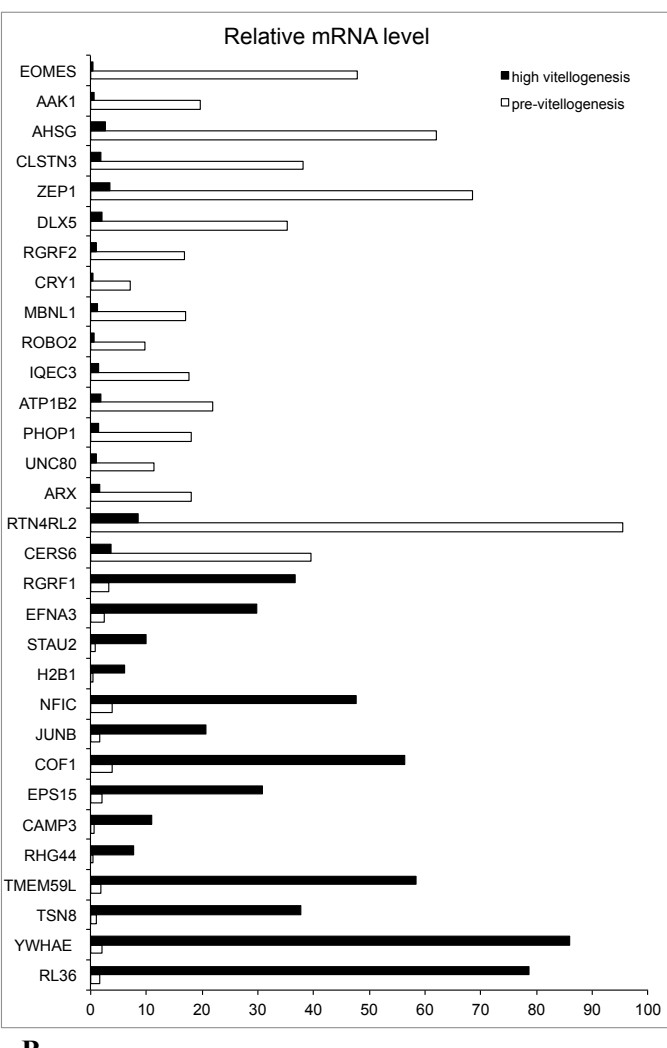

**B**

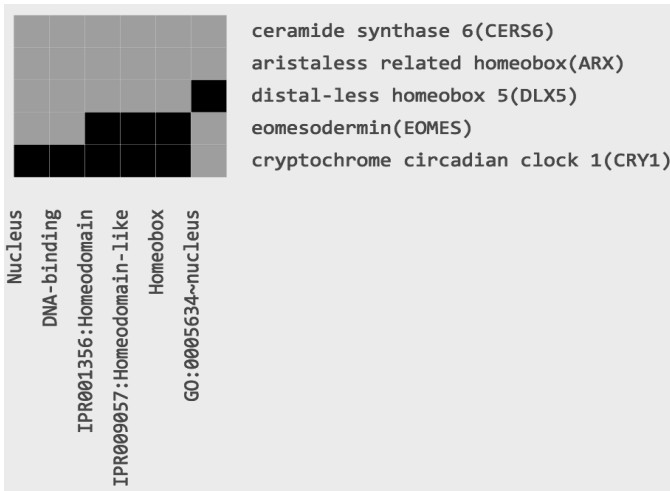

**Figure 2.** (**A**). Relative mRNA levels (based on RNA-Seq) of transcripts associated with known genes and showing significant differences in edgeR analysis (*P* < 0.001) between adult and juvenile blue gourami female brains. (**B**): Gene-annotation association of a cluster of genes encoding nuclear proteins, all upregulated in PVTL. Analysis was performed using DAVID software with default parameters. Grey tint of squares denotes a positively reported corresponding gene-term association.

Among the genes upregulated in PVTL compared to HVTL, a cluster of five genes encoding proteins localized to the nucleus, three of which contained the homeobox domain (ARX, DLX5, CERS6), was identified by functional annotation and DAVID clustering (Figure 2B). Additionally, 18 and 19 transcripts that could not be annotated by similarity to other species were marked as being significantly ($P < 0.001$) upregulated in PVTL and HVTL, respectively (data not shown).

In addition to a hypothesis-free analysis of genes with the greatest changes in expression levels, we identified changes in the transcript levels of the following genes with known involvement in reproduction: gonadotropin-releasing hormone II receptor (GnRHR), pituitary adenylate cyclase-activating polypeptide type I receptor (PACAP 1R), mPR, and GPCR (Figure 3A) (Sequences in Supplementary Table S2).

**A**

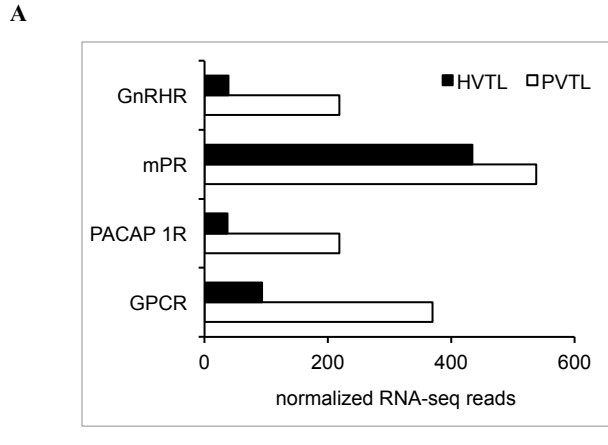

**B**

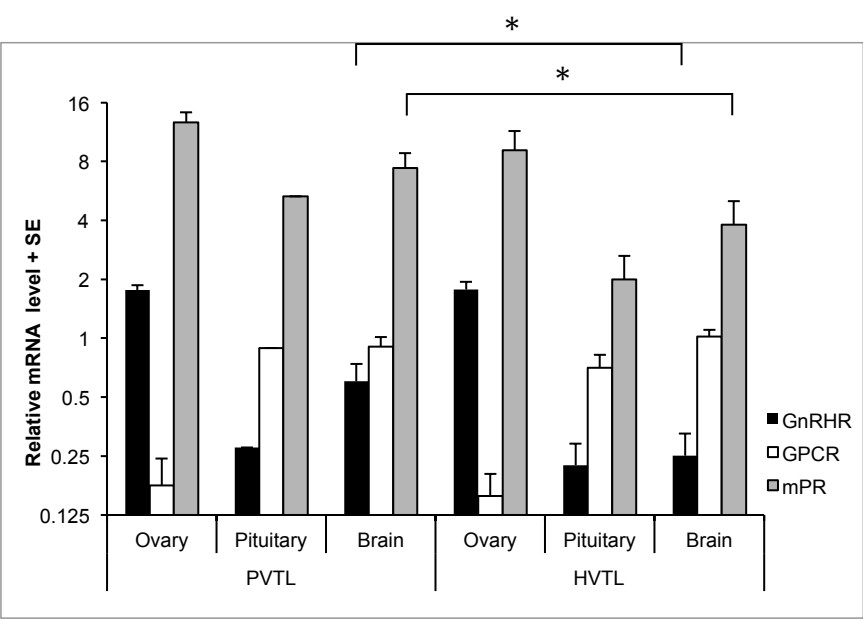

**Figure 3.** (**A**). mRNA levels (based on normalized RNA-seq reads) of gonadotropin-releasing hormone II receptor (GnRH IIR), pituitary adenylate cyclase-activating polypeptide type I receptor (PACAP 1R), membrane progesterone receptor epsilon (mPR) and G-protein coupled receptor 54 (GPCR) in the brains of blue gourami females at different stages of oogenesis: PVTL and HVTL. (**B**). Relative mRNA levels (based on qRT-PCR) of GnRH IIR, mPR, and GPCR in the brains of blue gourami females at different stages of oogenesis: PVTL and HVTL. Bars: S.E. from n = 5. *: $P < 0.05$.

*2.2. qRT-PCR*

The mRNA levels of GPCR, mPR, and GnRHR were further measured by qRT-PCR in the brain, pituitary, and ovary of individual fish in the PVTL and HVTL stages (n = 5 in each group). The average mRNA level of GPCR was ~4-fold lower in the ovary than in both the brain and pituitary, in both PVTL and HVTL stages ($P = 7 \times 10^{-5}$ and $P = 7 \times 10^{-6}$, respectively). Conversely, the transcription of GnRHR was ~7-fold higher in the ovary than in the brain and pituitary, in both PVTL and HVTL stages ($P = 3 \times 10^{-4}$ and $P = 4 \times 10^{-5}$, respectively). The average GnRHR and mPR expression in the brain was ~2-fold higher in PVTL than HVTL, with significance of $P = 0.032$ and $P = 0.045$ respectively (Figure 3B). No significant differences between the expression levels of these receptors between PVTL and HVTL were found in the pituitary or ovary. Likewise, no significant differences between the two stages of vitellogenesis were found in the mRNA level of mPR in any of the tissues (Figure 3B).

## 3. Discussion

In fish like in other vertebrates, the brain plays a central role in coordinating sexual function, including the regulation of oogenesis. In the present study, of the 34,368 unique transcripts identified, 23,710 could be annotated by similarity to genes in other species, and 69 differentially expressed transcripts were identified in the brain at PVTL and HVTL in blue gourami. In the brains of mature zebrafish, 825-1,046 genes associated with vitellogenesis were expressed differentially during the early stages of vitellogenesis [24], while in males, 1,828 differentially expressed transcripts were regulated by $E_2$. In the present study, only two developmental stages (PVTL and HVTL) were examined, while additional stages were examined in zebrafish. The involvement of BPG in the control of oogenesis has been described [1] and is supported by some of the results of the present study. Furthermore, our results support the notion of dimorphic patterns of gene expression in the brain of a sexually mature vertebrate model, with implications for studies on reproduction [25].

In our present findings, both RNA-seq and qRT-PCR showed that GPCR, GnRH IIR and mPR receptors in the brain were upregulated in PVTL compared to HVTL. These results are in partial agreement with the findings in [13], which showed high expression of GPCR, as well of the binding hormone Kiss2, in HVTL compared to PVTL.

During oogenesis, the levels of the mediating hormone GnRH1 [7] were high in HVTL compared to PVTL [13]. However, during vitellogenesis, mRNA levels of both Kiss2 and GnRH1 changed, but those of RnRH2, RnRH3 and Kiss1R did not. Kiss2R changed but was lower than Kiss2 between PVTL and HVTL, and no significant difference was found in the mRNA levels of the two stages [13]. GnRH1 is an important hormone controlling vitellogenesis in blue gourami [7]. Nakajo et al. [26] proposed that in non-mammalian vertebrates, especially teleosts, the existence of kisspeptin regulation on the HPG axis is still controversial. They suggested that in medaka and goldfish, Kiss does not control the BPG axis, and [27] reported that the kiss/kissr systems are dispensable for zebrafish reproduction.

A hypothesis explaining this result [13] is that Kiss 2 controls hypothalamus hormones by receptors of the GnRH [11,12,24,28]. It controls the release of pituitary gonadotropins, the FSH and the LH, which in turn control gametogenesis [1,2,7,8].

Additionally, our findings point to the downregulation of several transcription factors, including those containing the homeobox domain, in the vitellogenesis process. Their putative function as effectors and/or facilitators of the hormonal signaling described above requires further investigation.

In this study, we used pooled samples to obtain a general profile of changes in gene expression between two distinct developmental stages. In blue gourami, many genes associated with hormones involved in reproduction were studied using qRT-PCR (review, [1]). In these methods, the aim is limited to the study of one or more genes. In the present study, we sought to examine a large number of genes. Therefore, we used the brain transcriptome in the female blue gourami, which is a model for Anabantidae fish, during VTL. In this study, 69 differentially expressed transcripts were identified in the brain at PVTL and HVTL in blue gourami. Additionally, we examined several receptor genes important for reproduction, and the corresponding hormones which were described in previous studies

in blue gourami, e.g., GnRH II [7], PACAP [8] and Kiss 2 [13], as well as in other fish species. These results enable examining the expression of these receptors during vitellogenesis, as was done in the present study, and comparing the expression during oogenesis of those hormones as measured in previous studies (for a review, see [1]). In the present study, it was found that the mRNA levels of GnRH IIR, mPR, and GPCR were higher in PVTL than in HVTL, which agrees with their involvement in regulating vitellogenesis (Scheme, Figure 4). The GnRH IIR is a receptor of GnRH II whose mRNA levels did not change during the reproductive cycle, which is in agreement with several studies (for a review, see [8]). We observed that the mRNA level of GnRH IIR was dramatically higher in the ovary than in brain and pituitary in both stages of vitellogenesis. Additionally, during HVTL the ovary GnRH IIR mRNA level was significant higher than in PVTL. This supports the hypothesis that GnRH affects not only the pituitary but also the ovary via its highly expressed receptor.

In conclusion, this study presents changes in blue gourami brain transcriptome through gonadal development and vitellogenesis, supporting its central role in reproductive control. Our findings suggest that dozens of genes are involved in vitellogenesis, while only a few were previously described [1,13]; these RNA-seq findings require qRT-PCR validation in future studies, especially since the use of pooled samples could introduce a bias. Further studies are needed to characterize the specific genes involved in the brain control of vitellogenesis and their functions.

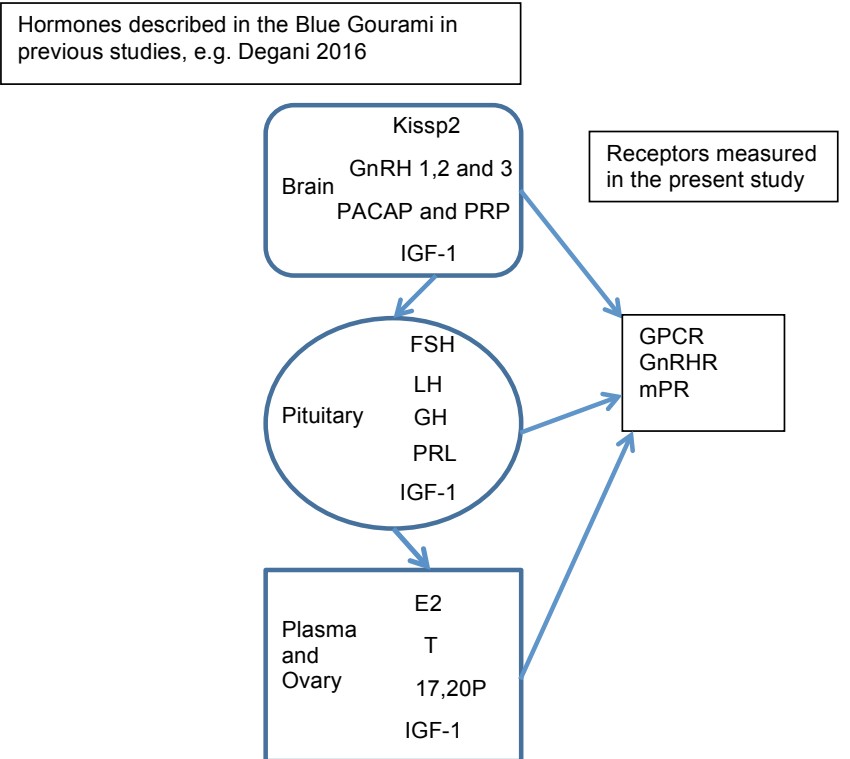

**Figure 4.** BPG axis hormones involved in regulating oogenesis in Anabantidae fish, as previously described in blue gourami, e.g., [1] and receptors measured in the present study In the brain: Kisspeptin 2-Kissp2, GnRH-gonadotropin-releasing hormones, pituitary adenylate cyclase-activating polypeptide (PACAP), and its related peptide (PRP) insulin-like growth factor-1 (IGF-1). In the pituitary: Follicle-stimulating hormone (FSH), luteinizing hormone (LH), growth hormone (GH), Prolactin (PRL). In the plasma and ovary: 17β-estradiol (E2), testosterone (T), 17α,20β-dihydroxy-4-pregnen-3-one (17,20P).

## 4. Materials and Methods

The general scheme of the study is based on our previously published work [29].

## 4.1. Fish and Sampling Procedure

Blue gourami fish (*T. trichopterus*) maintained and bred on the Ma'abarot fish farm on Kibbutz Ma'abarot, Israel, were used in this study. Investigations were conducted under the supervision of an in-house veterinarian, in accordance with the International Guiding Principles for Biomedical Research Involving Animals, as promulgated by the Society for the Study of Reproduction. The fish were grown in containers ($2 \times 2 \times 0.5$ m) at a temperature of 27 °C under a light regime of −12 h light/12 h darkness [19] and fed an artificial diet (45% protein, 7% fat) supplemented by live food (*Artemia salina*).

For the study of gene expression in the brain, pituitary, and ovary, two-month-old females at the PVTL stage of gonadal development (n = 5, 2.6 ± 0.57 g) and four-month-old females at the HVTL stage (n = 5, 4.6 ± 0.57 g) [17] underwent histological examination using light microscopy (characteristic morphology of gonadal cells presented in Figure 1B). The fish were anaesthetized with 0.03% tricaine methane sulfonate (MS222, Sigma-Aldrich), and their fork length and body weight were measured. Brains, pituitaries and ovaries were removed after decapitation as in [7], and brains were sectioned sagitally along the hemi-brains and frozen in 1.5 ml tubes with RNALater (Ambion) at −25 °C until further analysis.

## 4.2. Histological Analysis

Gonadal samples from females at different stages of gonadal development, as described previously (PVTL and HVTL), were fixed in Bouin and subsequently processed for light microscopy. Paraffin sections of 6 mm were stained with hematoxylin and eosin, as previously described [17] (Figure 1B).

## 4.3. RNA Extraction

Samples of five brains of juvenile females found in PVTL and five brains belonging to mature females found in HVTL were removed from RNALater (Thermo-Fisher Scientific) and homogenized using the TissueRuptor (Qiagen). Total RNA was extracted from each sample using TRI Reagent (Sigma) according to the manufacturer's protocol. The concentration and integrity of RNA were examined using a Thermo-Fisher Scientific NanoDrop 8000 Spectrophotometer and Agilent 2100 Bioanalyzer. All RNA samples had OD260/280 ≥ 1.8 and RNA integrity number (RIN) ≥ 7. Equal amounts of the five RNA samples from each group were then pooled together for cDNA synthesis and sequencing. RNA extraction from the pituitary and ovary was similarly performed.

## 4.4. Library Construction, Illumina Sequencing and Transcriptome Assembly

RNA-Seq library preparation and sequencing were carried out at the Genomics Center of the Silberman Institute of Life Sciences, Hebrew University of Jerusalem. cDNA libraries were prepared with ~2.5 μg of total RNA using NEBNext Ultra RNA library prep kit (New England Biolabs). The libraries were sequenced with one lane on an Illumina NextSeq 500 instrument with 75-bp single-end reads. The raw read files have been deposited in the NCBI Sequence Read Archive (SRA) with the accession PRJNA560182. Adaptor-only reads and low-quality reads were filtered out in the Illumina BaseSpace environment; the files were then examined using FastQC software (www.bioinformatics.babraham.ac.uk/projects/fastqc/). Cleaned reads were used for de novo assembly with Trinity software [30,31] with Kallisto transcript abundance estimation [32] and Bowtie alignment [33].

## 4.5. Functional Annotation

The assembly of RNA-Seq contigs was used for open reading frame extraction by TransDecoder [31] and similarity search as follows: BLASTX and BLASTP programs [34] with E value cutoffs of default and 1e-5, respectively, against the SwissProt/UniProt protein database [35]; and HMMSCAN program against the PFAM domain database [36]. Trinotate software [31] was used to integrate the hits into a single SQLITE database and generate an annotation report.

## 4.6. Gene Expression and Differentially Expressed Genes

Trinity software was used with the edgeR [37] method to identify differentially expressed genes between two libraries. The dispersion value was set at 0.01. DAVID online tools [38,39] (https://david.ncifcrf.gov) were used to find clusters of genes with similar functions in the differentially regulated gene list.

## 4.7. Quantitative RT-PCR

To validate the differential abundance of transcript for candidate genes based on RNA-seq results as well as prior knowledge, reverse transcription, primer design, and quantitative PCR were performed using SYBR Green chemistry and DNA primers. Primer sequences were designed using primer3 software based on transcriptomic sequences (Supplementary Table S2) and are provided in Supplementary Table S3. All primers were tested for efficiency (by serial dilutions) and specificity (by melting peak analysis). Reverse transcription was performed on an Applied Biosystems ABI-9600 with reagents from New England Biolabs. The qPCR was performed in technical quadruplicates on an Applied Biosystems ABI-7900HT Sequence Detection System equipped with a 384-well block. Data were analyzed using SDS 2.3 software (Applied Biosystems) and Microsoft Excel. Relative quantification and the ΔCq method were used. Results were normalized to the transcript abundance median of all measured genes per sample.

## 4.8. Statistical Analysis

The significance of the differences between the experimental groups, as quantified by RNA-Seq, was estimated by the edgeR differential expression pipeline [37,40]. Significance of qRT-PCR results was estimated by a two-tailed Student's t-test. Differences were considered statistically significant at $P < 0.05$.

**Supplementary Materials:** The following are available online at http://www.mdpi.com/2410-3888/4/4/54/s1: Table S1: Genes differentially expressed in PVTL/HVTL (as in Figure 2A) and their putative functions. Table S2: Blue gourami receptor gene sequence. Table S3: Blue gourami receptor gene primer sequences.

**Author Contributions:** Conceptualization, G.D. and A.M.; methodology, G.D. and A.M.; investigation, G.D., A.A., A.H., and A.M.; writing—original draft preparation, G.D. and A.M.; writing—review and editing, G.D. and A.M.; visualization, G.D. and A.M.; supervision, G.D. and A.M.; project administration, G.D.; funding acquisition, G.D. and A.M.

**Funding:** This research received no external funding.

**Acknowledgments:** The authors would like to thank H. Yehuda (MIGAL) for expert technical assistance.

**Conflicts of Interest:** The authors declare no conflict of interest.

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
