# Peer review of "Vitellogenesis in Blue Gourami is Accompanied by Brain Transcriptome Changes"

_fishes, doi:10.3390/fishes4040054_

Round 1

Reviewer 1 Report

The manuscript entitled: "Vitellogenesis in the blue gourami is accompanied by brain transcriptome changes" is well written and provide new insight into transcriptome changes connected with fish reproductive physiology. I found only minor issues which could be taken into account during preparation of the final version of manuscript.

Figure 1 – correct the descriptions on the figure. In recent format it is hard to read.

Figure 2 – describe the X-axis

Figure 3 – For what stands the question mark?

Line 154 – please check this phrase: “agreement with the findings in [13], which”. For reference see the sentence from Line 163-165.

Line 179 180– sentence: “In this study, 69 differentially high differently expressed transcripts were identified in the brain at PVTL and HVTL in blue gourami.” Need to be corrected.

Line 180 – 183 – Please recheck the sentence “Additionally, we examined several receptor genes important for reproduction, and the corresponding hormones (which)? were described in previous studies in blue gourami, e.g., GnRH II [7], PACAP [8] and Kiss 2 [13], as well as in other fish species.”

Line 185 and 189– indicating the literature as follow: “(review, [1])” could be replaced with” (for a review, see [1])”.

References:

No 19 - Check the reference, no information regarding the source provided.

Author Response

We thank the Reviewer for the positive assessment and the detailed comments which have enabled us to correct several errors in the manuscript.

In the revised file, Figure 1A is enlarged to improve visibility following formatting. In Figure 2, the X-axis description ("Relative mRNA level") was moved to under the axis to improve clarity. We were not able to locate a question mark anywhere in the manuscript, it was probably the result of a formatting error in the pdf conversion process. We have corrected all the other errors indicated by the Reviewer, and updated the references as requested.

Reviewer 2 Report

The aim of this study was to identify the blue gourami brain transcriptome in pre-vitellogenesis and high vitellogenesis that control yolk accumulation in ovogenesis. From 34,368 unique transcripts identify by RNA sequencing of whole brain, the authors was focused on the transcripts showing significant differences between stages. They also investigated mRNA levels by quantitative RT-PCR in brain, pituitary and ovary for three receptors involved in hormonal control of vitellogenesis. This study helps to identify key genes involved in fine mechanism of blue gourami ovogenesis.

Author Response

We thank the Reviewer for the positive assessment of our manuscript!